# Talent Selection Based on Sport-Specific Tasks Is Affected by the Relative Age Effects among Adolescent Handball Players

**DOI:** 10.3390/ijerph182111418

**Published:** 2021-10-29

**Authors:** Zsófia Tróznai, Katinka Utczás, Júlia Pápai, Zalán Négele, István Juhász, Tamás Szabó, Leonidas Petridis

**Affiliations:** 1Research Centre for Sport Physiology, University of Physical Education, 1123 Budapest, Hungary; utczas.katinka@tf.hu (K.U.); szabo.tamas@mksz.hu (T.S.); petridis.leonidas@tf.hu (L.P.); 2Independent Researcher, 8651 Balatonszabadi, Hungary; papaijulia7@gmail.com; 3Teleki Blanka High School and Primary School, 8000 Székesfehérvár, Hungary; negelezalan@gmail.com; 4Hungarian Handball Federation, 1087 Budapest, Hungary; juhasz.istvan@mksz.hu

**Keywords:** youth sport, birth asymmetry, maturity, adolescence

## Abstract

Talent selection is often affected by the relative age effects (RAEs), resulting in the overrepresentation of relatively older (vs. relatively younger) players among those selected. The use of sport-specific tasks is suggested to reduce RAEs during talent selection. Purpose: To test the hypothesis that talent selection including only sport-specific tasks is not affected by the RAEs and to analyse the body size and biological maturity of the top selection level according to relative age. Methods: Participants were U14 female (*n* = 5428) and U15 (*n* = 4408) male handball players participating in four programs consisting of four selection levels (local, county, regional, and national) grouped in bi-annual age. Handball-specific generic skills, position-specific technical drills and in-game performance were the selection criteria evaluated by experts and coaches. Body dimensions were measured and bone age, as an indicator of maturity, was estimated. The relative age quartile distributions within the bi-annual cohorts were examined using Chi-square and Odds Ratios. Results: In terms of all the registered players no RAEs were evident. However, the RAEs of moderate effect size were evident at the county level; *χ*^2^ = 53.2 (girls) and 66.4 (boys), OR = 2.5 and 3.3, respectively. The RAEs of a large effect size were found at the regional level; *χ*^2^ = 139.5 (girls) and 144.9 (boys), OR = 8.2 and 5.2, respectively. At national level, RAEs were still present, but with no further increase in the effect size. At the highest selection level, there were no differences in the anthropometric measures between the relatively older and younger players. Conclusion: The findings provide support to the hypothesis that the selection process exacerbates RAEs even when using only sport-specific selection criteria. The performance metrics in technical skills, but also coaching assessments are likely involved. In addition, an advanced maturity and/or an above-average body size increases the selection odds for relatively younger players.

## 1. Introduction

Relative age effects (RAEs) are a well-known but relevant factor in youth sport [1,2,3]. During the developmental years, RAEs are described as an indirect factor [4], which may affect training and competition opportunities and potentially the athletes’ long-term careers [5]. According to RAEs, athletes born early in a selection period (e.g., a calendar year) tend to be overrepresented in age-group competitions compared to those born later [6].

From a sports perspective, Wattie et al. [4] proposed a theoretical approach for understanding RAEs phenomenon, which is based on Newell’s triangular framework of interacting constraints [7]. Three types of constraints are considered in this framework [4]: individual constraints, task constraints, and environmental constraints. Individual constraints include individual qualities, such as body size or maturity status. Task constraints include the demands and determinants of a specific sport. In handball for instance, where body size and physical attributes are key elements for success [8,9], relatively older athletes may be probably more advantaged over relatively younger. Finally, environmental constraints include policies and the broader socio-cultural environment [4]. The age grouping method, the competition system or the structure of talent identification process are typical elements classified here.

In line with the hypothesis of task constraints in handball, studies have revealed significant RAEs in several countries [3,10,11,12,13]. RAEs show a higher prevalence predominantly during mid and late adolescence [12,14], yet, significant birth asymmetries have been reported during the early adult years, mainly at an elite level. For instance, Wrang et al. [10] and Bjorndal et al. [11] reported an overrepresentation of the relatively older players in Danish and Norwegian international handball teams, respectively. Similar findings were reported also by Rubia et al. [3], who demonstrated that the RAEs affected performance at international competitions; the relatively older players played more and achieved a better performance compared to the relatively younger players.

The research has identified the depth of competition and the talent identification process as facilitators of the RAEs [15,16,17]; however, with mixed results in handball. For example, Schorer et al. [12] demonstrated an overrepresentation of relatively older handball players at higher competition levels within the national handball system in Germany. On the contrary, Wrang et al. [10] noted that, with the increase in competition level and in the age of the players, the effect size of RAEs decreased. Regarding the talent selection, Camacho-Cardenosa et al. [13] found significant differences among U17 players selected for the last Spanish Regional Championship for both men and women. On the other hand, Schorer et al. [18] reported no differences in the birth distributions between players selected for the national team and those not selected; however, there were still significant RAEs compared to the normal distribution in both groups.

The RAEs were related to differences in maturation between early- and late-born athletes [19], which potentially could result in selection bias in favour of the more matured. To reduce the effects of maturity in talent identification, Matthys et al. [20] proposed the use of sport-specific tasks for selection purposes in handball. Using the peak height velocity as an overall indicator of maturity, the authors studied how maturity affects performance in technical, sport-specific and physical testing tasks in young handball players [20]. The results revealed that maturation and body size seem not to influence performance in sport-specific skills, whereas performance in generic physical tests was clearly affected by the players’ maturity status [20]. The use of sport-specific tasks in talent selection is also supported by findings suggesting that adolescence sport-specific skills, such as ball handling and dribbling, are a more efficient discriminating factor between the elite and non-elite handball players than performance in simple physical tests [21]. However, the use of sport-specific tasks in the talent selection has not been yet studied experimentally. The question emerging here is how independent the sport-specific skills are from the RAEs when used in the talent selection.

Within a new, multi-level talent selection process, organised for one single age group of adolescent male and female handball players, the purpose of this study was to test the hypothesis that talent selection, including only sport-specific tasks, is not affected by the relative age effects. A secondary purpose was to analyse the body size characteristics and biological maturity at the top selection level according to the players’ relative age. Our hypothesis was that the birth distributions among the selected players will demonstrate limited, if any, evidence of RAEs. An additional important aspect was also to examine whether it is the selection process, which generates RAEs or whetherRAEs are already prevalent at the basic level of all registered players, indicating that the birthdate asymmetry appears in younger age groups.

## 2. Materials and Methods

### 2.1. Participants

The study included four selection programs, two for all registered U14 female handball players (*n* = 2562; mean age ± *SD*: 13.3 ± 0.6 years and *n* = 2866; mean age ± *SD*: 13.3 ± 0.6 years), and two for all registered U15 male handball players (*n* = 2111; mean age ± *SD*: 14.3 ± 0.6 years and *n* = 2297; mean age ± *SD*: 14.3 ± 0.6 years). Participants were grouped in bi-annual age categories based on the year of birth: from 1 January 2002 to 31 December 2003 (selection#1) and from 1 January 2004 to 31 December 2005 (selection#2) (Table 1). In order to be included in the analysis, players had to be registered members of Hungarian handball clubs and had to hold a valid license to participate in official regional or national competitions. The official records of the Hungarian Handball Federation provided the birth data of the players, and the Hungarian Central Statistical Office birth data of the corresponding Hungarian population (girls and boys born from 2002–2003 and from 2004–2005).

### 2.2. The Selection Programs

The selection programs were organised and supervised by the Hungarian Handball Federation. Four selections were examined in this study, two for female and two for male players. All four programs had the same structure, consisting of four selection levels, starting from the local (club) level, and ending with the selection for the national team. The selection levels were:

Local (club) level: From the pool of all registered players (13–14 years old girls and 14–15 years old boys), based on the recommendations of the team coaches, players were selected to participate at the next selection level, which included sport-specific criteria.

County level: Handball-specific generic skills, position-specific technical drills and in-game performance were the selection criteria. A detailed description and illustration of the selection tasks is provided in the Appendix A. Briefly, handball-specific skills consisted of two tasks, the same for all players, performed under timed conditions: one task for defensive skills (defensive footwork), and one task for offensive skills, which required dribbling the ball first in a straight line and then performing a jump-shot; and then dribbling along a zig-zag line, again ending with a jump-shot on goal. The time duration of the footwork task was evaluated (maximum 5 points), and technique (maximum 1 point) and time duration (maximum 4 points) for the dribbling–shooting task. Position-specific drills included typical moves and technical tasks according to playing positions: two tasks for backcourts (maximum 10–10 points), two for the wings (maximum 10–10 points), one for the pivots (maximum 20 points), and one for goalkeepers (maximum 20 points). Experts and coaches evaluated passing and shooting accuracy, goal scoring, and technical execution. In-game performance included the evaluation of offensive and defensive movements (5–5 points, respectively). The players with the highest scores were selected for the next (regional) level.

Regional level: Players had to perform only the position-specific technical drills under time pressure. At this level there was no rest between attempts; the tasks had to be performed at a higher intensity thereby, increasing the level of difficulty. Evaluation and scoring was the same as for the county level. Additionally, the in-game performance was again evaluated. The same experts as on the county level administered and supervised the measurements. Players with the highest scores were selected for the next (national) level.

National level: At this level, the evaluation included only in-game performance. Based on the scores a final ranking by playing position was determined. The players selected at this level went through additional measurements (anthropometric and physical tests), and the results were used to define the training development needs for each player individually.

### 2.3. Anthropometry and Biological Status

Body weight of the players who participated at the national selection level was measured with a Seca digital scale to the nearest 0.1 kg. Body height was measured with an anthropometer (DKSH Switzerland Ltd., Zürich, Switzerland) to the nearest 0.1 cm according to the recommendations of the International Standards for Anthropometric Assessment [22]. Biological age was estimated from bone age with an ultrasound-based device (Sunlight Medical Ltd., Tel Aviv, Israel). This method estimates the developmental stage of the skeleton and was shown to have high a reliability in boys up to 16 years and in girls up to 15 years [23]. Although the examination covers only a few bones, it can give a comprehensive assessment of the development of the skeletal system and can be used as an overall indicator of biological development [24,25]. We performed measurements on the wrist region of the left hand. Participants put their arms on the armrest surface between the transducers. We adjusted the transducers to the forearm (radius and ulna) growth zone, at the connection of the distal epiphysis and diaphysis. Then, at the initial position the device connected to the forearm at a pressure of approximately 500 g and emitted ultrasound at a frequency of 750 kHz to the measurement site for each measurement cycle. One measurement cycle lasted about 20 s and was repeated five times. The device estimates bone age (in years and months) using equations based on the speed of ultrasound and the distance between the transducers. Based on the difference between chronological and bone age the participants’ maturity status was estimated. The same person performed all measurements in during the morning. Inclusion criteria required a selection at the top (national) selection level and an informed consent given to participate in anthropometry and bone age measurements. Players were excluded, when they reported illness, pain or injury, especially in the wrist region within one year prior to the measurements, or failed to give informed consent. Due to injury or illness, we excluded from the measurements six female and five male players from the first selection and two female and twelve male players from the second selection.

### 2.4. Statistical Analysis

We examined relative age effects based on the month of birth in quarter-year intervals (from Q1 to Q8). We used a Chi-square test (*χ*^2^) to examine the difference between the observed and expected distributions for each selection level and we calculated an Odds Ratios (OR) from Q1 to Q8 between consecutive selection levels. The results for the anthropometric variables and biological status measured at the top selection level represented means ± *SD*. Differences in mean values between the two selection programs per gender were examined using Hedge’s *g* standardized effect sizes. The magnitude of the difference in evaluating the between-groups effect sizes was [26] trivial (<0.2), small (≥0.2), moderate (≥0.5), and large (≥0.8). For a more concise analysis of body size and maturity in relation to relative age, we grouped the athletes in half-year intervals (from S1 to S4) and used a one-way analysis of variance (with a Tukey post hoc test) for intergroup differences (S1–S4) for boys and girls separately. We used SPSS 25.0 (IBM) for the statistical analysis.

## 3. Results

The birth date distributions for the two selection programs for boys and girls separately were similar. We merged the two selection programs according to gender and presented the results accordingly. The distribution between the registered handball players and the corresponding Hungarian population was significant, but with no evidence for RAEs. For all next selection levels, we compared the relative age distribution to that of the registered players. For both female and male players, the distribution at the county, regional, and national levels differed significantly from that of the registered players (Figure 1 and Figure 2, Table 2). Q1 had the highest representation, while Q8 had the lowest. When the distributions were compared to the preceding level, the RAEs increased significantly until they reached the regional level. From the regional to the national level, the RAEs did not differ in either selection program. The odds ratios between Q1 to Q8 increased gradually from the registered players to the regional level and then decreased at the national level.

Players selected at the national level participated in anthropometric and bone age measurements. We compared the mean values of body size and maturity between the first and the second selection programs for both female and male players at the national level. The differences were trivial (body height and body weight: effect sizes: <0.2) to small (decimal age, bone age: effect sizes: 0.2–0.3). Then, we merged the two selection programs for girls and for boys and compared the S1 to S4 groups. There were no differences in anthropometric or bone age measures between the semester groups, except for the bone age for the female players in the S4 group, which was significantly lower compared to the other semesters (Table 3).

## 4. Discussion

In this study, we examined the relative age effects within four similar, multi-level talent selection programs for adolescent female and male handball players. A unique characteristic of the examined talent selection programs was that the selection was based on only handball-specific tasks. The main question, therefore, was to analyse whether the sport-specific selection criteria could reduce or remove the relative age effects from the selection process.

With the aim to select the most talented players, the Hungarian Handball Federation designed a multi-level talent selection process, organised for one single age group: U14 female and U15 male players. This is the first age group with official national-wide competitions. At the end of this program, 60 players were selected at the top (elite) level and then the federation coaches selected the most talented to represent the U15 national team.

The main finding of this study was that the talent selection for adolescent handball players was affected by RAEs in both girls and boys, even when using only sport-specific selection criteria. The selection bias is further magnified by the bi-annual age grouping method. The players born in the second year had about five to six times fewer chances to be selected at the top level. Since the selection program was organized every two years, these players were excluded from any further selection across the following age-groups, decreasing their odds for an athletic career at top level.

The results, however, should be examined according to the selection levels starting from the level of all registered players. At this level, we compared the players’ birth distribution to that of the corresponding average population examining whether an uneven birth distribution already existed among all the registered players. According to some earlier studies, the relative age effects appeared in early adolescence (11–12 years) in sports such as soccer, handball and volleyball [14]. Delorme and Raspaud [27] have reported RAEs at even younger ages (~7 years) among basketball players. These studies suggest that a birthdate asymmetry already exists among all the registered players and as described by Delorme and Raspaud [27], RAEs are the result of self-elimination at the beginning of young athletes’ athletic career and are not generated by any selecting or recruiting methods. The beginning of RAEs at the level of all registered players was not confirmed. The birth distribution of registered players was inconsistent, in some cases even with a higher representation of relatively younger, indicating the absence of RAEs (Figure 1 and Figure 2). Overall, the results do not support the idea of an already existing asymmetry at the level of all the registered players. They rather underline the significant contribution of the selection process to the appearance of the RAEs.

RAEs for all programs, which were significant but of moderate effect size, were found at the first selection level (from local to county level). For both female and male players, relatively older players were consistently overrepresented and had more than two times (girls) and more than three times (boys) higher chances of being selected than the relatively younger players. The consistency in the birth asymmetry through all four programs suggests a systemic selection bias in favour of the relatively older players. Since the selection at this level was based on the recommendations of the coaches and team managers, these findings emphasize the central role of the coaches in the appearance of RAEs. The selection of talented players in team sports is often connected to the subjective opinion of the coaches or scouts [28]. However, and most likely due to the need of the coaches to have immediate success in age group competitions, the coaches may more easily select players with larger body dimensions and/or advanced maturity [29]. Not surprisingly, the teams with the relatively older players are usually more successful than teams with relatively younger players [30]. An emerging argument within RAEs research is the need to educate and raise the attention of the coaches to the effects of relative age [2,31]. This is supported by findings where RAEs decreased when the coaches were aware of the players’ relative age [28]. However, there was one study [32], where the coaching awareness of RAEs during talent selection of 12–15 years soccer players did not remove nor reduced it.

From county to regional level and onwards, the selection criteria included performance evaluation in sport-specific, position-specific technical tasks and in-game performance. The implementation of sport-specific tasks during the selection process was suggested to provide an evaluation of the players’ abilities independently of maturity status [18,20]. In contrast to previous findings [20] and to our expectations, RAEs significantly increased from the county to regional level in both female and male players. On average, the players born in the Q1 group had six (males) to eight (females) times more chances of being selected. At this point, it was assumed that by excluding physical attributes from the selection criteria we could roughly equalize the selection chances between the relatively older and younger players. It should be noted, however, that the exclusion of physical attributes was only partially achieved, particularly at the handball-specific generic skills. For example, the rapid execution of the defensive footwork task, but also successful execution of the jump-shots required, to a large extent, a developed physique. Additionally, the evaluation of the in-game performance may also be biased by the RAEs, with relatively older players more likely to dominate over relatively younger players on the field. Collectively, the results did not confirm our assumption and, as supported previously [12,16], indicated that the progression towards the elite level increased the magnitude of RAEs, even when the selection was based solely on sport-specific criteria.

The RAEs from regional to national level did not change significantly but, compared to all the registered players, were still present and significant. This was because the relative age distribution followed a similar pattern that was generated at previous levels. This was mostly visible in the male players, who showed a significant decrease in the effect size from 5.15 to 1.61 (Table 2), implying that at the national level relative age does not seem to affect the selection process. The relative age at the top selection level had a higher impact in female players with the odds ratio being almost unchanged (>7) compared to the regional level (Table 2).

A second purpose of this study was to analyse the body dimensions and biological maturity status at the national level, which represents the elite level in the selection program. Out of all the registered players, about 2.5% reached this level and, from those players, about half of them became members of the age group national teams participating in international competitions. Regarding body dimensions, two main observations can be established: first, the superior body size of the handball players at an elite level, which is in line with previous studies [33]. When compared to the normative data [34], the elite female players in our sample were taller by 7.8 cm and heavier by 8.6 kg than girls of the same age (14.5 years) from the general population and were above the 90th and 75th percentile for height and weight, respectively. Similarly, the elite male players in this study were taller by 7.6 cm and heavier by 8.8 kg, compared to normative data from the general population (15.5 years boys) [34] and were above the 75th percentile for both height and weight.

A second observation was the homogeneity in body dimensions at the top level independent of the players’ relative age. The consistent lack of differences in all four programs between semester groups clearly indicate the importance of these attributes for elite performance. These findings confirm earlier studies stating that players at the elite level tended to have similar anthropometric and physique characteristics [35]. This may also explain the plateau in RAEs experienced at regional level. It is possible, that players not complying with the size and physique requirements were excluded from the selection at earlier levels irrespective of their relative age.

The differences between chronological age and bone age suggest an advanced maturity at the top selection level with a larger effect size in male (1.8 years) vs. female (0.9 years) players. This is a well-known phenomenon in youth sport [36] and again emphasizes the advantage of early maturity during the selection. It is noteworthy that, despite the differences in relative age, the bone age did not differ between the semester groups except in a very few cases in the S4 group for girls. This means that, compared to their chronological age, the relatively younger players were more biologically advanced than the relatively older players (Table 3). Such results indicate that, primarily in male players, maturity may have a larger impact on selection than the relative age itself and shows that, among the relatively younger advanced maturity increased selection chances. It should be mentioned that in this study we used bone age only in relation to the players’ chronological age to estimate maturity status. Furthermore, irrespectively of maturity, the very few cases of relatively younger female players at the top selection level with superior body size, but without early maturity (Table 3), indicate an independent significant effect of body size. However, this is limited by the very small sample size and needs to be examined in a larger sample and at lower selection levels.

A main limitation in this study was that the present data did not allow the anthropometric measures and the biological age of selected and non-selected players at the lower selection levels to be examined (registered players, county, and regional levels). In future research, such data could help to better understand the impact of the relative age, maturity, and body dimensions on the selection process, independently of each other, and how homogenization occurs towards the elite level. Additionally, we did not examine the RAEs according to the playing positions. It is possible, that due to differences in the body size requirements (e.g., backcourts vs. wings), the magnitude of RAEs would differ between playing positions.

## 5. Conclusions

The findings confirm the connection of talent identification for adolescent female and male handball players with the RAEs. A novel approach in this study was that the selection process included only sport-specific selection tasks. The selection criteria were the same for both female and male players, allowing for a separate analysis of RAEs. The findings provide support the following hypotheses: (i) the early stages of the selection process, even when using only sport-specific selection criteria, increase the RAEs; (ii) although of moderate effect size, the coaches’ recommendations may play a key role in the appearance of RAEs; and (iii) superior body size and advanced maturity increase the selection chances for relatively younger players.

The results question the necessity of selection at this age range, irrespective of the selection criteria. Governing bodies should consider interventions in the structure of the talent selection or in the age-grouping methods to reduce the impact of relative age. Additionally, relative age and maturity should be included in performance evaluations, even when using technical tasks. Future research should examine the applicability of performance corrective adjustments according to the relative age and maturity. Furthermore, the quantification of the RAEs according to playing positions and to the profile of the selection tasks could give a deeper understanding of the RAEs in youth handball. Finally, selection programs should be more flexible, offering developmental opportunities for unselected athletes and ensuring different pathways to the elite level for athletes of a different relative age and/or maturity status.

## Figures and Tables

**Figure 1 ijerph-18-11418-f001:**
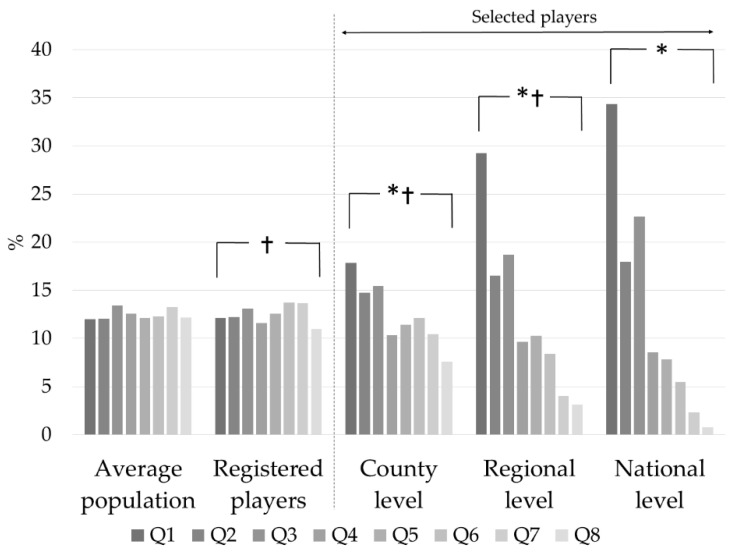
Relative birth distribution of all U14 female handball players and of the selected players (county, regional, and national levels). Q1–Q8: Annual quartiles; Average population: girls of the same age born in Hungary. *: *p* < 0.05 compared to all registered players, †: *p* < 0.05 compared to the preceding level.

**Figure 2 ijerph-18-11418-f002:**
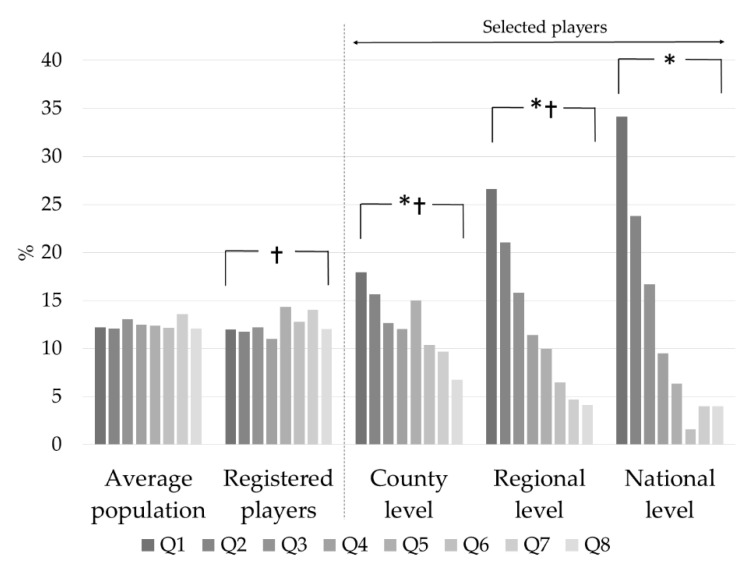
Relative birth distribution of all U15 male handball players and of the selected players (county, regional, and national levels). Q1–Q8: Annual quartiles; Average population: boys of the same age born in Hungary. *: *p* < 0.05 compared to all registered players, †: *p* < 0.05 compared to the preceding level.

**Table 1 ijerph-18-11418-t001:** Participants at each selection level for the four selection programs.

	Selection#1 (Females)	Selection#2 (Females)	Selection#1 (Males)	Selection#2 (Males)
Selection period	2016	2018	2017	2019
Birth years	2002–2003	2004–2005	2002–2003	2004–2005
Average population (*n*)	92,893	93,528	98,558	99,105
Registered players (*n*)	2562	2866	2111	2297
County level (*n*)	474	434	326	448
Regional level (*n*)	130	191	152	190
National level (*n*)	63	65	66	60

Average population: boys and girls of the same age born in Hungary.

**Table 2 ijerph-18-11418-t002:** The odds ratios and chi-square results according to the selection level for U14 female and U15 male handball players.

**Female Players**	**Average Population** **(*n* = 186,421)**	**Registered Players** **(*n* = 5428)**	**County Level** **(*n* = 908)**	**Regional Level** **(*n* = 321)**	**National Level** **(*n* = 128)**
*χ*^2^ to registered players			53.2 *	139.5 *	98.6 *
*χ*^2^ to preceding level		23.1 †	53.2 †	51.7.9 †	7.8
OR (Q1 vs. Q8)			2.49	8.16	7.92
**Male Players**	**Average Population** **(*n* = 197,663)**	**Registered Players** **(*n* = 4408)**	**County Level** **(*n* = 774)**	**Regional Level** **(*n* = 342)**	**National Level** **(*n* = 126)**
*χ*^2^ to registered players			66.4 *	144.9 *	103.6 *
*χ*^2^ to preceding level		26.4 †	66.4 †	46.6 †	10.0
OR (Q1 vs. Q8)			3.29	5.15	1.61

Q1–Q8: Annual trimesters; OR = odds ratio. Average population: girls and boys of the same age born in Hungary. *: *p* < 0.05 compared to all registered players, †: *p* < 0.05 compared to the preceding level.

**Table 3 ijerph-18-11418-t003:** Descriptive results of anthropometric and biological age assessments of the players participated at the national selection level (mean ± SD).

**Variables**	**Female Players**	**S1**	**S2**	**S3**	**S4**
*n*	121	66	36	16	3
%	100	54.2	30.0	13.3	2.5
CA (year)	14.3 ± 0.4	14.6 ± 0.2	14.2 ± 0.2 *	13.6 ± 0.1 †	13.1 ± 0.2 ^‡^
BA (year)	15.2 ± 1.1	15.4 ± 1.0	15.1 ± 1.2	14.8 ± 1.3	13.6 ± 0.2 ^•^
BA-CA (year)	0.9 ± 1.1	0.8 ± 1.0	0.9 ± 1.2	1.2 ± 1.2	0.5 ± 0.3
Body Mass (kg)	62.8 ± 8.4	63.0 ± 7.8	63.5 ± 10.2	60.2 ± 6.6	63.9 ± 5.0
Body Height (m)	1.71 ± 0.07	1.71 ± 0.06	1.71 ± 0.07	1.68 ± 0.07	1.73 ± 0.05
**Variables**	**Male Players**	**S1**	**S2**	**S3**	**S4**
*n*	109	63	28	8	10
%	100	57.8	25.7	7.3	9.2
CA (year)	15.3 ± 0.5	15.6 ± 0.1	15.1 ± 0.1 *	14.6 ± 0.2 †	14.0 ± 0.1 ^‡^
BA (year)	17.1 ± 0.9	17.1 ± 0.9	17.2 ± 0.9	16.8 ± 1.2	16.7 ± 0.7
BA-CA (year)	1.8 ± 1.0	1.5 ± 1.0	2.1 ± 0.9	2.1 ± 1.2	2.6 ± 0.7
Body Mass (kg)	73.6 ± 11.3	74.3 ± 11.9	74.1 ± 10.0	74.8 ± 13.9	66.4 ± 7.4
Body Height (m)	1.82 ± 0.07	1.83 ± 0.08	1.82 ± 0.06	1.81 ± 0.03	1.79 ± 0.09

S1–S4: Annual semesters; CA: Chronological Age; BA: Bone Age. *: *p* < 0.05: S2 vs. S1, S3, S4. †: *p* < 0.05: S3 vs. S1, S2, S4. ^‡^: *p* < 0.05: S4 vs. S1, S2, S3. ^•^: *p* < 0.05: S4 vs. S1.

## Data Availability

The data presented in this study are available upon request from the corresponding author. A video material with all tasks is also available upon request from the corresponding author.

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
