# Peer review of "Talent Selection Based on Sport-Specific Tasks Is Affected by the Relative Age Effects among Adolescent Handball Players"

_ijerph, 2021, doi:10.3390/ijerph182111418_

Round 1

Reviewer 1 Report

The present study gives insights about the impact of RAE/body maturity/dimensions on the selection of young Hungarian handball players. The straight of this study is the number of the involved players, which are representative for the young handball population. The same side, I would suggest the author to start with a preliminary study in order to better defines the protocol of assessment, and to choose which selection could be the more interesting to discuss. 

The abstract doesn't provide some necessary results, and considers the topic generally.

Introduction: the RAE concept is not sufficiently explained and the framework must be deeper detailed. From line 62 to 75 the contents should be put in the discussion and not in the introduction. The aims should be explained with more clearness and details. All the assumption should be placed in the materials and methods.

Methods: the paragraph 2.2. doesn't provide precise data about the selection criteria (for instance by saying "position-specific technical tasks", reproducing the tasks is impossible).

Inclusion and exclusion criteria should be explained properly. 

Results: the tables are too many and are creating confusion. The authors should focus on the most important results and create some tables for those; the same, they should discuss other data in the text. Like this, it's unreadable.

Discussion: in this chapter the authors should speak about the results, by re-citing them in the text. It's necessary to highlight some p-values or some OR in order to make the text more comprehensible and readable. There are many speculations which are not supported by any literature, and the impact of the present results and the importance of the study should be highlight.

Author Response

We would like to sincerely thank the reviewer for the time to review this manuscript and for the valuable comments, which improve the manuscript’s quality. Please find our response to the comments below. Corrections in the revised manuscript appear in Track changes function.

The present study gives insights about the impact of RAE/body maturity/dimensions on the selection of young Hungarian handball players. The straight of this study is the number of the involved players, which are representative for the young handball population. The same side, I would suggest the author to start with a preliminary study in order to better defines the protocol of assessment, and to choose which selection could be the more interesting to discuss. 

Response: We thank the reviewer for this comment. We reconsidered statistical analysis and the presentation of the results trying to make them easier to follow and more understandable. The results of the four selection programs were quite similar, which indicates a systemic and consistent phenomenon of the examined selection programs. We considered to globally interpret the results of all programs together. Our intention in this way, was to point out the most important outcomes and conclusions.

The abstract doesn't provide some necessary results, and considers the topic generally.

Response: We agree with this comment, we revised the abstract and added some more details about the methods and the most important results, also taking into consideration the limitations in words number.

Introduction: the RAE concept is not sufficiently explained and the framework must be deeper detailed. From line 62 to 75 the contents should be put in the discussion and not in the introduction. The aims should be explained with more clearness and details. All the assumption should be placed in the materials and methods.

Response: We revised the introduction and tried to address the reviewer’s comments. We added a separate paragraph including the theoretical framework for understanding the relative age effects (RAEs), as this was described by Wattie et al. (2015). This framework uses the triangular Newell’s theory of interacting constraints and in sports context defines the three constraints that influence the prevalence of the RAEs and need to be considered in such research. As proposed by Wattie et al. (2015) the constraints are individual constraints, task constraints, and environmental constraints. A description was added in the introduction.

As suggested, we moved the contents previously being under lines 62-75 to the discussion.

We rephrased the aims trying to make the purpose of the study clearer. Hypothesis was also rephrased and follows the aims of the study, we considered not to move hypothesis from this paragraph following the journal’s general instructions, which suggest that hypothesis should be included in the introduction.

Nick Wattie, Jörg Schorer, Joseph Baker. The Relative Afe Effect in Sport: A Developmental Systems Model. Sports Med (2015) 45:83-94

Methods: the paragraph 2.2. doesn't provide precise data about the selection criteria (for instance by saying "position-specific technical tasks", reproducing the tasks is impossible).

Response: We agree with this comment, and we understand that reproduction of the task based on the provided information was not possible. Two main aspects were considered here: first, the very detailed description of the selection tasks arising from their complexity and their variety according to playing positions. Second, the nature of technical tasks, which makes them difficult to standardise. We added a detailed description with illustrations of the tasks and the evaluation aspects as a supplementary file accessible to anyone. We decided not to include all these information in the manuscript, we believe such a description would be hard to follow and perhaps could cause confusion. Hopefully, based on the description provided as a supplementary material, the tasks became clearer and can also be reproduced. A video material with all tasks has been also prepared and is available upon request.

Inclusion and exclusion criteria should be explained properly. 

Response: For the RAEs analysis, players had to be registered players of any handball club and had to hold a valid licence to participate in official competitions. This information was rephrased and can be found under Materials and methods/participants (lines: 105-107). For the anthropometry and maturity analysis at the top selection level inclusion criteria required an informed consent to participate in anthropometry and bone age measurements, while players were excluded, when they reported illness, pain or injury, especially on the wrist region within one year prior to the measurements or failed to give informed consent. We added this information under Materials and methods/Anthropometry and biological status (lines: 166-170).

Results: the tables are too many and are creating confusion. The authors should focus on the most important results and create some tables for those; the same, they should discuss other data in the text. Like this, it's unreadable.

Response: Yes, we fully understand the complexity of the tables. We revised statistical analysis and restructured the presentation of the results. To make it clearer we merged the two selection programs per gender and presented the results accordingly. Also, we used figures for a more concise view of the distributions, and we prepared a new table, which contains the results of the statistical analysis with the values of the chi-square test and the most important odds ratio. We believe that the figures more clearly demonstrate differences in relative birth distributions at the basic level (average population and registered players) and at the different selection levels.

Discussion: in this chapter the authors should speak about the results, by re-citing them in the text. It's necessary to highlight some p-values or some OR in order to make the text more comprehensible and readable. There are many speculations which are not supported by any literature, and the impact of the present results and the importance of the study should be highlight.

Response: We revised parts of the discussion and tried to highlight and discuss the most important results. As suggested, we refer to OR values in the discussion (for example: lines 269-270, 291-292, 305-306) or to results from the anthropometric measurements using also reports from the literature. Additionally, we removed those parts that contained speculations, we agree their added value was limited. We added a paragraph under conclusions trying to summarize the impact and the practical applications of the study making also suggestions for some possible interventions that could be made from policy makers to reduce the impact of RAEs on talent selection, and for future research.

Reviewer 2 Report

The manuscript presents a very important and relevant topic for sport. The RAE is very studied and still presents very controversial results, which requires even more studies in the area, with different variables that can influence this phenomenon. However, the manuscript needs changes.
General: The manuscript needs to present more information about RAE. In addition, the analysis should also be performed by quartiles (4 trimesters), separately between the age-groups.
Specific: ABSTRACT
I suggest reducing the first part of the abstract and writing more about the methods, mainly about the programs and the selection criteria used by them, in a very brief and objective way.
The results are not presented clearly, there is little information.
KEYWORDS
Remove numbers after words.
INTRODUCTION
The authors do not present the theoretical models that support the RAE (eg, Hancock et al, 2013; Wittie et al., 2015; and others), which leaves the introduction without a clear explanation of the phenomenon. I suggest that the authors better describe the RAE, with its theories and studies.
The effects of RAE in handball according to age-groups and different levels, including those related to talent selection, are well described in the literature. However, the authors present few studies to justify the present work, not making clear the lack in the literature related to the subject. I suggest that the authors present more data related to this lack, presenting the current state-of-the-art in works with the RAE in handball.
Finally, present clear hypotheses for the study.
METHODS Participants: I suggest add the mean (and SD) age of the groups Selection programs: I suggest add information from the instruments used for the assessments, specifically at County, Regional, and National levels. Anthropometry and biological status: Add references to the procedures used. Statistical analysis: I suggest redoing the analyzes by quartiles in each age-group. DISCUSSION I suggest writing a separate paragraph with study limitations and suggestions for further studies. Include a paragraph with practical applications of the study with possible solutions for the results found.  

Author Response

We would like to sincerely thank the reviewer for the time to review this manuscript and for the valuable comments, which improve the manuscript’s quality. Please find our response to the comments below. Corrections in the revised manuscript appear in Track changes function.

Comments and Suggestions for Authors

The manuscript presents a very important and relevant topic for sport. The RAE is very studied and still presents very controversial results, which requires even more studies in the area, with different variables that can influence this phenomenon. However, the manuscript needs changes.
General: The manuscript needs to present more information about RAE. In addition, the analysis should also be performed by quartiles (4 trimesters), separately between the age-groups.

Response: We thank the reviewer for this comment. We revised the introduction to give a better description of the RAEs, mainly in handball research. Also, as suggested, we reconducted statistical analysis using quartiles and presented the results accordingly.

Specific: ABSTRACT
I suggest reducing the first part of the abstract and writing more about the methods, mainly about the programs and the selection criteria used by them, in a very brief and objective way.
The results are not presented clearly, there is little information. 

Response: We revised the abstract and added more details about the methods and the most important results. However, we had to also consider the limitations in words number.

KEYWORDS
Remove numbers after words.

Response: corrected accordingly

INTRODUCTION
The authors do not present the theoretical models that support the RAE (eg, Hancock et al, 2013; Wittie et al., 2015; and others), which leaves the introduction without a clear explanation of the phenomenon. I suggest that the authors better describe the RAE, with its theories and studies.
The effects of RAE in handball according to age-groups and different levels, including those related to talent selection, are well described in the literature. However, the authors present few studies to justify the present work, not making clear the lack in the literature related to the subject. I suggest that the authors present more data related to this lack, presenting the current state-of-the-art in works with the RAE in handball.

Finally, present clear hypotheses for the study.

Response: We revised the introduction as suggested and we added a separate paragraph including the theoretical framework for understanding the relative age effects (RAEs). We used the framework as described in the work of Wattie et al. (2015), which is based on the triangular Newell’s theory of interacting constraints. Wattie et al. (2015) places this theory into sports context and defines the three constraints that influence the prevalence of the RAEs and need to be considered in such research. As proposed by Wattie et al. (2015) the constraints are individual constraints, task constraints, and environmental constraints. Also, we tried to improve the presentation of the current state of research in summarizing the most important publications studying the RAEs in handball. These new and revised contents can be found under introduction (lines 53-61, 62-87).

Nick Wattie, Jörg Schorer, Joseph Baker. The Relative Age Effect in Sport: A Developmental Systems Model. Sports Med (2015) 45:83-94

METHODS Participants: I suggest add the mean (and SD) age of the groups Selection programs: I suggest add information from the instruments used for the assessments, specifically at County, Regional, and National levels. Anthropometry and biological status: Add references to the procedures used. Statistical analysis: I suggest redoing the analyzes by quartiles in each age-group.

Response: we added mean±SD values for the participants of each selection program. This information can be found under Methods and materials/partcipants (lines 102-104).

We added a detailed description with illustrations of the selection tasks and the evaluation aspects as a supplementary file accessible to anyone. We refer to this material in the manuscript under Methods and materials /The selection programs/County level (lines 119-120). In the manuscript we revised the description of the tasks, however we decided not to include all these information in the manuscript due to the complexity and their variety of the tasks according to playing positions. We believe such a description in the manuscript would be hard to follow and perhaps could cause confusion. Hopefully, based on the description provided as a supplementary material, the tasks became clearer and can also be reproduced. A video material with all tasks has been also prepared and is available upon request. In the manuscript, we revised this part, attempting to give a more comprehensive idea of the selection process.

As suggested, we added references for the anthropometric measurements and also, we extended the description of bone age measurements and provided references as well to support this method.

As suggested, we revised the statistical analysis by using quartiles. We also restructured the presentation of the results. We removed previously tables 2 and 3, we merged the two selection programs per gender and prepared two figures to demonstrate relative birth distributions at the basic level (average population and registered players) and at the different selection levels. Also, we prepared a new table, which contains the results of the statistical analysis with the values of the chi-square test and the most important odds ratio.

DISCUSSION I suggest writing a separate paragraph with study limitations and suggestions for further studies. Include a paragraph with practical applications of the study with possible solutions for the results found.  

Response: In the revised manuscript we tried to better highlight the main limitation of the study, which can be found under discussion (lines 358-366). Also, and as suggested, we added a paragraph under conclusions trying to summarize the impact and the practical applications of the study, summarizing some possible interventions that could be made from policy makers to reduce the impact of the RAEs on talent selection. Suggestions for future research are mentioned under the last paragraph of the discussion (lines 346-349) and of the conclusions (lines 362-363).

Round 2

Reviewer 1 Report

Please, try to highlight the conclusions related to the study itself and to your results more than to your opinion. Thanks

Author Response

We would like to thank once more the reviewer for the comments contributing to the improvement of the manuscript. 

During this revision, we made some corrections in the text, as suggested, mainly in the Discussion and Conclusions paragraphs. All corrections were made with the track changes funcion in MS word. We tried to more clearly describe the results of this study and emphasize the most important outputs. In the Conclusions, we summarized the three main conclusions that can be made based on the results. More specifically, that during the selection RAEs increased, even when using only sport-specific selection tasks, referring also to the key role of the team coaches in this process. Our third conclusion was related to the importance of body size and maturity at the top selection level. We understand, that in this study we did not examine the independent effects of the relative age, body size and maturity on the selection, it is our pleasure to undertake such analysis in future research.

Also, the manuscript went through English language editing and proofreading.

Reviewer 2 Report

The authors made substantial changes to the manuscript, meeting most considerations and considerably improving understanding.

Author Response

We would like to thank the reviewer for the kind comments regarding the corrections we made during the first revision.

During this revision, we made some minor corrections in the Methods. Hopefully, after both revisions and with the supplementary file describing the selection tasks, study methods became easier to understand and even reproduce.

We express once more our gratitude for the review and the valuable comments contributing to improving the overall quality of the manuscript.